# Crystallinity and Play-of-Colour in Gem Opal with Digit Patterns from Wegel Tena, Ethiopia

**Kehan Zhao and Feng Bai ***

School of Gemmology, China University of Geosciences (Beijing), Beijing 100083, China;
2109180006@cugb.edu.cn
**\*** Correspondence: baifeng@cugb.edu.cn

**Abstract:** A typical feature of Wegel Tena opal is the "digit pattern". This pattern consists of two parts, columns and matrix, with different colours, transparency or play-of-colour effect, which is still unexplained. This study aims at investigating the various parts of the digit pattern using different spectroscopic methods, and scanning and transmission electron microscopy (SEM and TEM). The band at 780 cm$^{-1}$ on the Fourier transform infrared (FTIR) spectrum is correlated to the symmetric stretching vibration of Si–O. The bands at 1085, 895, 785 and 3600 cm$^{-1}$ on Raman spectra indicate that Wegel Tena opal is opal-CT. Comparison of the relative intensity of the Raman signals around 360 cm$^{-1}$ indicates that the microcrystalline opal on the top of the sample contains a higher amount of tridymite-like structural units, and the tridymite-type regions in the matrix contain a higher degree of structural defects. Silica spheres in the columns tend to be smaller and better ordered than in the matrix. The diameter of the silica spheres ($d$ = 80–500 nm) or agglomerates ($d$ = 200–580 nm) in Wegel Tena opal satisfies the conditions of diffraction of visible light, and the thickness of the silica layer ($h$ = 120–200 nm) satisfies the conditions for film interference.

**Keywords:** Ethiopia; opal; digit patterns; play-of-colour; FTIR; Raman spectroscopy; SEM; TEM; opal-CT

## 1. Introduction

Opal is an amorphous hydrated silica to micro-crystalline silica mineral, with a chemical formula of $SiO_2 \cdot nH_2O$ [1,2]. According to Jones and Segnit, opals are divided into opal-C, opal-CT and opal-A, based on X-ray diffraction (XRD). Opal-C is a relatively well-ordered $\alpha$-cristobalite with minor evidence of tridymite; opal-CT is a disordered $\alpha$-cristobalite with $\alpha$-tridymite type stacking; opal-A is amorphous. In recent years, however, this classification has been questioned, and the nature of opal-CT remains far from conclusive [1]. Wilson explained the opal-CT structure by a tridymite-dominant paracrystalline model [3]. Recent work by Curtis et al. [4] found that opal-CT appears to be a paracrystalline form of silica that has some structural characteristics of cristobalite and tridymite.

Other techniques, such as infrared spectroscopy [5–7] and Raman spectroscopy [8–14], have also been used to distinguish different types of opal. Furthermore, Raman spectroscopy has been considered a method to distinguish the complex structural relationships between different types of opal [4,8,10,13].

Silica spheres can be observed in some of the opals using an electron microscope. When the size and arrangement of the spheres meet certain conditions, a special optical effect, play-of-colour, can appear [1,2,13]. Gem opal with play-of-colour is called "precious opal"; conversely, opal without play-of-colour is called "common opal". Play-of-colour is independent of the structural features revealed by XRD. The atomic structure of the silica in the spheres of opal-A and opal-CT is clearly different, although both of them can show play-of-colour [4]. Precious opals are mostly mined in Australia,

Mexico, Brazil and Ethiopia. In the early 1990s, opal was discovered in Mezezo, Shewa Province, Ethiopia. The body colour was mostly orange to red or "chocolate" brown [15–18]. In 2008, high-quality precious opal was discovered in Wegel Tena, in the northeast of Wollo Province, Ethiopia [19–21]. The deposit occurs in an Oligocene volcano-sedimentary sequence of alternating basalt and rhyolitic ignimbrite layers [22]. Opals mined at Wegel Tena [18,21] and Mezezo [16,17,21] always show a "digit pattern", a macroscopic finger-like structure. This pattern consists of almost parallel vertical columns, separated by a homogeneous matrix of a different colour, transparency or play-of-colour [21,23]. Rondeau et al. [23] built a model of digit formation based on observation containing four basic steps: (1) The occurrence of sedimentation and dehydration of silica spheres, which take on a columnar structure; (2) polygonization of the columns; (3) water inflowing, altering the columns by erosion along the polygonal boundaries; (4) new silica-rich fluid filling the interstice and settling, the formation of silica lepispheres and dehydration.

This study analyses the crystallinity and microstructure in opals from Wegel Tena, finds the difference between columns and matrix, and explains the cause of play-of-colour in Wegel Tena opal by linking the microstructure with the various effects.

## 2. Materials and Methods

We studied 11 opals from Wegel Tena, Wollo Province, Ethiopia. Six of them were cabochon (Figure 1), while five were rough samples with surrounding rocks (Figure 2). Each sample showed play-of-colour. Opals 01, 02, 03, 04, 05, 06, 09 and 11 showed digit patterns. All cabochon opals had polished surfaces, and polishing had no effect on the spectral data.

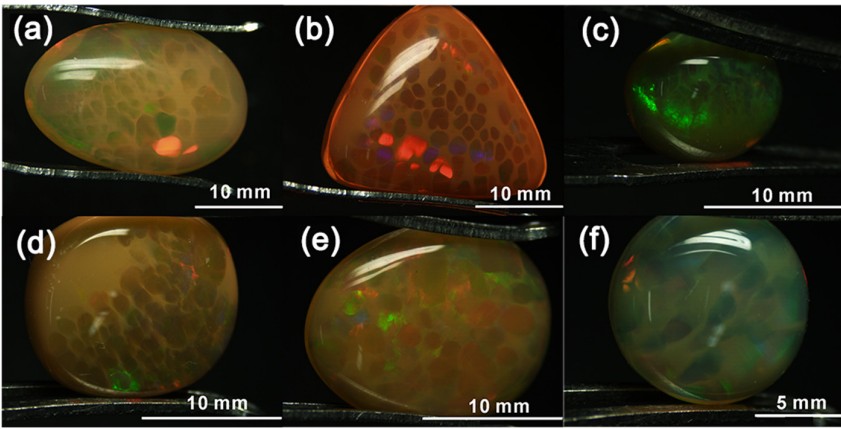

**Figure 1.** Cabochon samples. (**a**) Opal-01; (**b**) Opal-02; (**c**) Opal-03; (**d**) Opal-04; (**e**) Opal-05; (**f**) Opal-06.

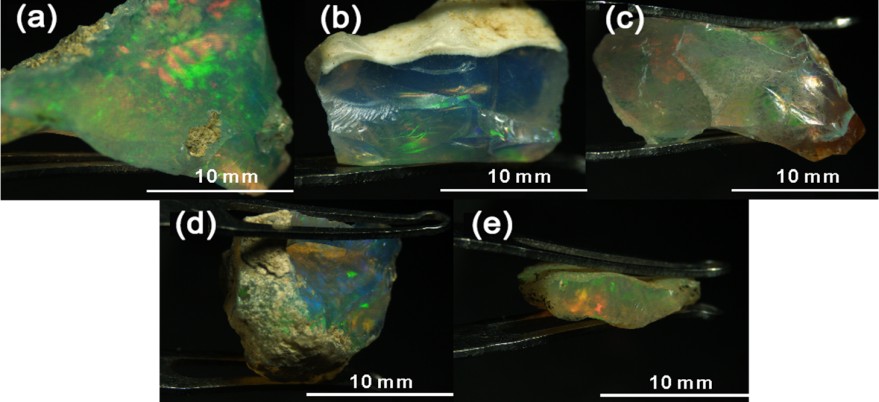

**Figure 2.** Rough samples with surrounding rock. (**a**) Opal-07; (**b**) Opal-08; (**c**) Opal-09; (**d**) Opal-10; (**e**) Opal-11.

We used a refractometer to measure the index of refraction (RI) with an optical contact liquid of 1.78 RI. Ultraviolet (UV) fluorescence was observed using a UV 240 shortwave (254 nm) and longwave (365 nm) UV lamp. Samples were observed using a gemmological microscope with a maximum magnification of 40, using reflected and transmitted light.

FTIR and Raman spectroscopy were used to identify the silica forms. These are non-destructive and do not require preparation of the samples [24–26]. The work was carried out at the Experimental Teaching Center of Jewellery and Mineral Materials, China University of Geosciences, Beijing.

Infrared spectra in the mid-infrared range (2000 to 400 cm$^{-1}$) were recorded for all 11 samples on a polished surface in reflection mode, using a Tensor 27 FTIR (Bruker, Ettlingen, Baden-Württemberg, Germany). Thirty scans were collected at a resolution of 4 cm$^{-1}$.

Micro-Raman spectra were measured on seven samples (Opals 01, 02, 03, 04, 05, 06 and 08) and collected in the 100 to 4000 wavenumber (cm$^{-1}$) region using a LabRAM HR Evolution confocal Raman spectrometer (Horiba, Kyoto, Kyoto-fu Japan), with a 600 lines/mm grating (500 nm blaze) and an excitation wavelength of 532 nm at a power of 30 mW. Each spectrum was scanned two times for 5 seconds, with a 100 μm slit and a 100 μm laser hole. ICS correction (instrument specific intensity correction) was on. Samples were observed using a 50x VIS LWD (visible and long working distance) objective (Horiba, Kyoto, Kyoto-fu, Japan). Two to four points were selected for each sample. For the samples with strong transparency contrast, the Raman spectra were collected in the matrix and columns respectively. All spectra were baseline corrected by LabSpec software (version 5). Baseline correction did not affect band positions and their relative intensity ratios. The broad band around 360 cm$^{-1}$ was selected from several spectra, and fitted with Voigt functions using Origin 9 software (version 9.0.0.45).

The micromorphology of opals was observed on surfaces and fresh-fractured material. Samples were coated with a thin Pt film. High-resolution images of six samples (Opals 01, 03, 04, 07, 08 and 09) were obtained with a SUPRA 55 Sapphire scanning electron microscope (SEM) at the Institute of Earth Science in China University of Geosciences (Beijing), equipped with a field-effect electron gun, at an acceleration voltage of 15 kV. The test temperature was 22 °C, and the humidity was 45%.

Three samples (Opal-01, 05, 06) were prepared into flakes by mechanical grinding and ion thinning to electron transparency. The test surface of the flake was vertical to the length of columns. The microstructure was investigated on an H-8100 transmission electron microscope (TEM) provided by the School of Earth Sciences and Resources of China University of Geosciences (Beijing). The accelerating voltage was 200 kV, and the observation magnification was from 900 to 10,000.

## 3. Results

### 3.1. Appearance and Gemmological Properties

The appearance and gemmological properties of the samples are presented in Table 1. The body colours are brown, yellowish-brown, orange-brown, yellow and colourless. The digit patterns can appear on samples of different colours and sizes. The columns and matrix are of similar hue on the same sample, but the transparency is different. Columns are transparent, and the matrices are translucent or semi-translucent. The refractive index (RI) values of the samples range from 1.430 to 1.465 and are typical for opal, but slightly higher than the data measured by Rondeau et al. [18]. Fluorescence under UV is white or bluish-white luminescence, and fluorescence under LWUV (longwave ultraviolet) is stronger than under SWUV (shortwave ultraviolet).

The cross-sectional diameter of columns is about 0.2 to 5 mm, and the shape is from angular polygonal to oval or round. The cross-sectional shape of columns tends to be similar on the same sample (e.g., Opal-02, see Figure 3), but large variations also occur (e.g., Opal-05, see Figure 4). In the samples with the characteristic digit patterns (e.g., Opal-02, as shown in Figure 3), the columns are parallel to each other, and the ends of the columns are rounded. According to the model of digit formation [23], the honeycomb-like side of Opal-02 is the bottom, and the round-end of the column covered by common opal is the top. In the sample Opal-09 with weak digit patterns (Figure 5), the matrix is dendritic.

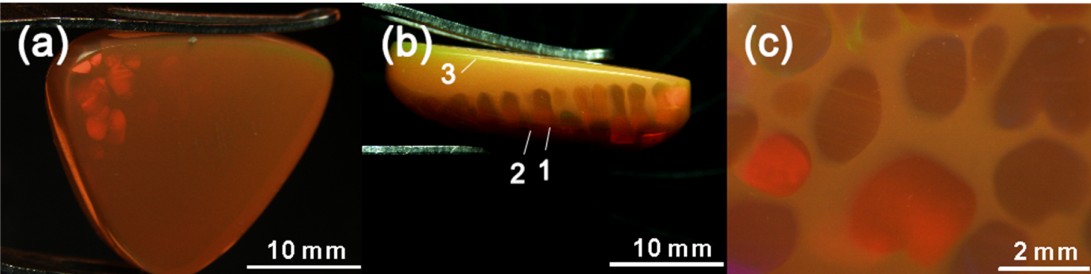

**Figure 3.** Opal-02 showing the characteristic digit pattern. Columns (1) are transparent with a vivid play-of-colour, separated by common opal matrix (2); (**a**) The top of the sample is covered by common opal (3); (**b**) When viewed from the perpendicular direction, the surface shows a finger-light structure. Transparent areas are the vertical sections of parallel columns and rounded at one end; (**c**) When viewed from the bottom of the sample, the surface shows a mosaic pattern. The transparent rounded patches are cross sections of columns, separated by an opaque matrix composed of common opal [23].

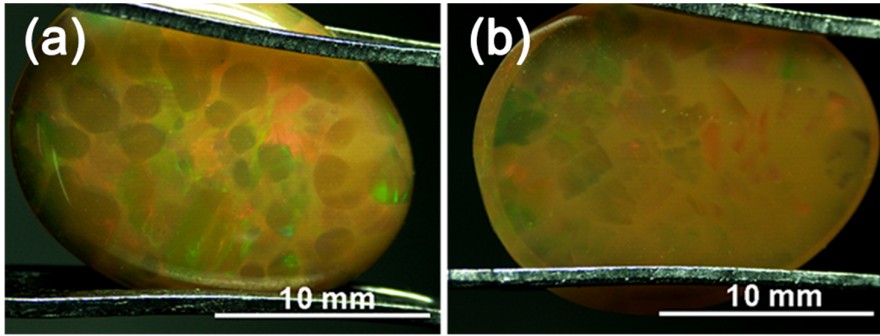

**Figure 4.** Play-of-colour is shown on both columns and matrix in Opal-05. The shape of the digits is very different. On the one side, (**a**) the cross section of the columns is near-circular, and on the other side (**b**) they are angular polygons.

**Table 1.** Descriptions of appearance and gemmological properties. Opals 07, 08 and 10 do not show a digit pattern. No visible inclusions are observed under the gemmological microscope.

| Sample | W [1] (ct) | Colour of Columns | Transparency of Columns | Colour of Matrix | Transparency of Matrix | RI [2] | LWUV [3] Fluorescence | SWUV [4] Fluorescence |
|---|---|---|---|---|---|---|---|---|
| Opal-01 | 3.369 | Pale-yellowish-brown | Transparent | Pale-brown | Translucent | 1.430 | White, moderate | White, weak |
| Opal-02 | 4.144 | Pale-orange-brown | Transparent | Orange-brown | Semi-translucent | 1.465 | White, weak | White, very weak |
| Opal-03 | 0.537 | Yellow | Transparent | Brown | Translucent | 1.464 | White, weak | White, very weak |
| Opal-04 | 1.017 | Pale-yellowish-brown | Transparent | Pale-brown | Semi-translucent | 1.459 | Inert | Inert |
| Opal-05 | 0.909 | Pale-yellowish-brown | Transparent | Pale-brown | Translucent | 1.464 | White, weak | White, very weak |
| Opal-06 | 0.659 | Pale-brown | Transparent | Brown | Translucent | 1.460 | White, weak | White, very weak |
| Opal-07 | 2.473 | Colourless | Transparent | - | - | N.M.[5] | Bluish-white, moderate | Bluish-white, weak |
| Opal-08 | 1.214 | Colourless | Transparent | - | - | N.M. | Bluish-white, moderate | Bluish-white, weak |
| Opal-09 | 2.898 | Pale-yellow | Transparent | Pale-brown | Translucent | N.M. | Inert | Inert |
| Opal-10 | 2.093 | Pale-yellow | Transparent | - | - | N.M. | Inert | Inert |
| Opal-11 | 0.625 | Pale-brown | Transparent | Pale-brown | Translucent | N.M. | Inert | Inert |

[1] Weight. [2] Index of refraction. [3] Longwave ultraviolet radiation. [4] Short wave ultraviolet radiation. [5] Not measured.

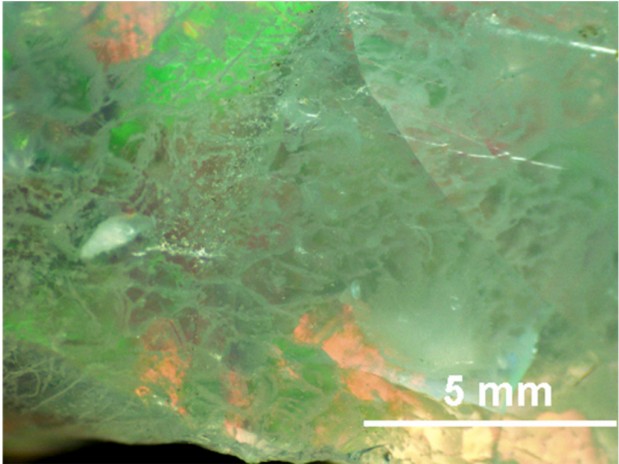

**Figure 5.** The digit pattern in Opal-09 is undeveloped and has a dendritic matrix.

The play-of-colour patches are bright and two-dimensional silk-like, flaming, flaky, or band-shaped, and show bright and dark stripes parallel to each other (Figure 6). A single patch usually consists of only one colour in one direction of observation, and a continuous spectral colour also appears. There are two kinds of spectral colours, one being the pure spectral colour with clear boundaries (Figure 7a,b), and the other being the spectral gradient colour with blurred boundaries (Figure 7c). In most samples (e.g., Opal-02, see Figure 3), play-of-colour appears only in columns, and the matrix is common opal. A patch can be divided into parts by the matrix, and on adjacent columns, the shape and texture of the colour patch are continuous (Figure 3). Sometimes, play-of-colour can be seen on both columns and matrix (Figure 4). A patch on columns can be divided by the matrix, and a patch on the matrix is divided by columns. The patches on these two types of areas are not connected to each other.

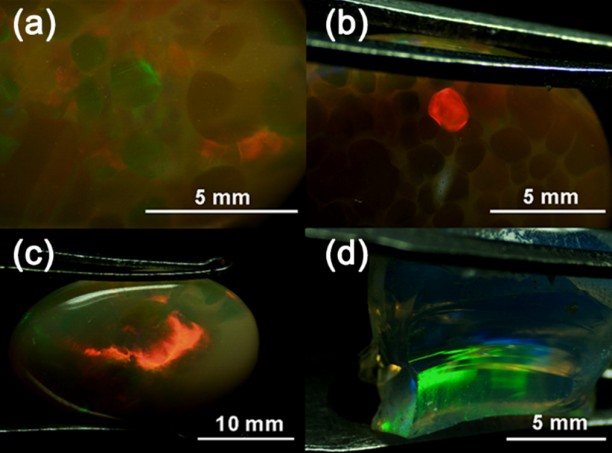

**Figure 6.** Different shaped play-of-colour patches in Ethiopian opals. (**a**) **S**ilk-like (Opal-05); (**b**) Flaky (Opal-04); (**c**) **F**laming (Opal-01); (**d**) Band-shaped (Opal-08).

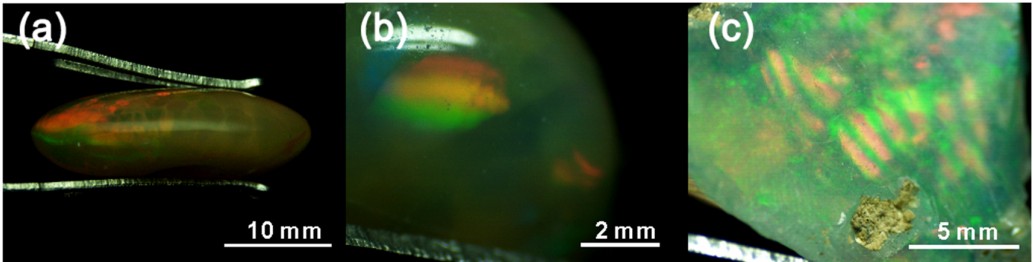

**Figure 7.** Spectral colour in Ethiopian opals. (**a**) Pure spectral colour (Opal-01); (**b**) Pure spectral colour (Opal-06); (**c**) Gradient spectral colour (Opal-07).

*3.2. FTIR Spectroscopy*

The infrared reflectance spectra of all samples are similar (Figure 8), with three strong bands around 474, 780 and 1100 cm$^{-1}$, relating to the fundamental Si–O vibrations. An inflexion point appears around 1245 cm$^{-1}$. The 1100 cm$^{-1}$ band is caused by the antisymmetric stretching vibration of Si–O ($\nu_{as}$ (Si–O)), and the 780 cm$^{-1}$ band is caused by the symmetric stretching vibration of Si–O ($\nu_s$ (Si–O)). The 474 cm$^{-1}$ band is related to O–Si–O bending vibration $\delta$ (O–Si–O) [5,27–33]. The transition from the amorphous to the crystalline form is evident in the FTIR spectra. One characteristic of this transition is a significant inflexion point at 1245 cm$^{-1}$ [34,35]. Opal-C can be identified by the absorption appearing at 480, 620, 675, 800 and 1110 cm$^{-1}$, especially the intense bands at about 630 cm$^{-1}$ [5–7]. The infrared spectra of our samples are obviously different from those of opal-C, and more similar to the spectra of opal-A and opal-CT. The symmetric stretching vibration of the Si–O absorption band in opal-CT is in the range of 788 to 792 cm$^{-1}$, while this band is at 796 to 800 cm$^{-1}$ in opal-A [5]. The band measured in this study is around 780 cm$^{-1}$, which is the closest to the opal-CT.

However, it is not convincing enough to rely on the small difference of band position at around 780 cm$^{-1}$. Due to the limitations of the FTIR test conditions (the size of the infrared spot is larger than most columns), the columns and matrix could not be tested separately as well. Therefore, we used a micro-Raman spectrometer to further study the fine-scale structure of samples.

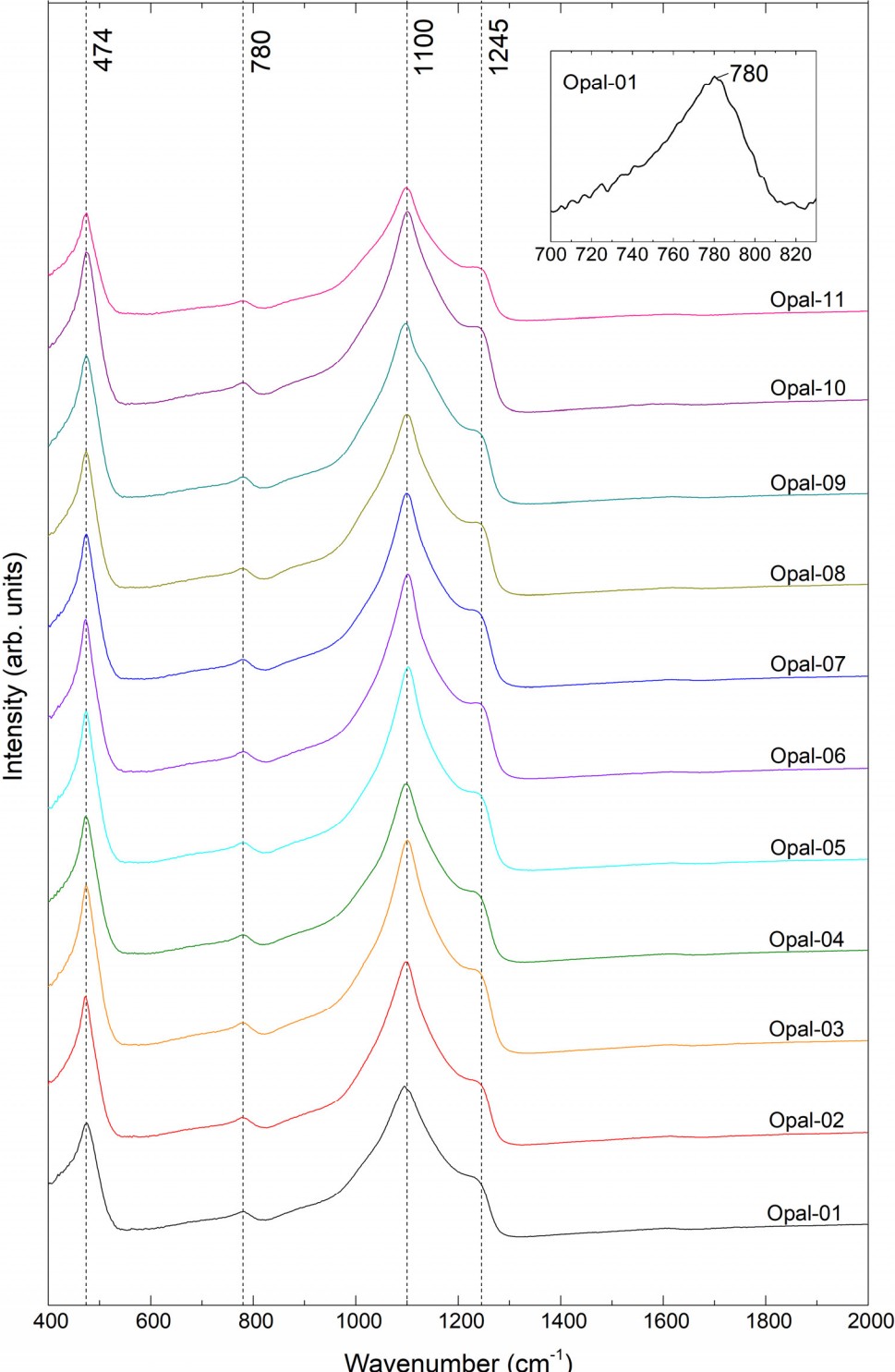

**Figure 8.** Infrared spectra of all 11 samples, with three bands around 474, 780 and 1100 cm⁻¹, and an inflexion point at 1245 cm⁻¹.

*3.3. Micro-Raman Spectroscopy*

Each sample shows a high Raman background, which is most likely related to the occurrence of crystalline defects interacting with light or the presence of water [36]. Raman spectra were baseline corrected to eliminate the influence. All samples show the following Raman shifts (Figure 9): broad

bands around 360 cm$^{-1}$, a series of weaker bands in the range of 750 to 1650 cm$^{-1}$, as well as broad bands around 2945 cm$^{-1}$ and 3550 cm$^{-1}$. Antisymmetric Si–O–Si stretching vibration bands are located around 1210 and 1085 cm$^{-1}$; the Si–O stretching vibration band is located around 895 cm$^{-1}$; the band located around 785 cm$^{-1}$ is assigned to symmetric Si–O–Si stretching vibration [10,13,37,38]. The bands from 1600 to 3600 cm$^{-1}$ are related to water: the band around 2945 cm$^{-1}$ indicates "cristobalite" water [39,40]; the band centred at 1610 cm$^{-1}$ and the broad band around 3550 cm$^{-1}$ are caused by physiosorbed water molecules [12,18,41]. Bands at 1085, 895, 785 and 360 cm$^{-1}$ are characteristic of opal-CT [8–13].

The broad band centred around 360 cm$^{-1}$ is mainly composed of five signals: cristobalite-type atomic arrangements at 230 and 420 cm$^{-1}$, tridymite-type nanoscale spatial regions at 300 and 355 cm$^{-1}$, and a shoulder around 490 cm$^{-1}$ representing vibration of Si–O–Si rings with three to four terms [12,13]. By analysing this broad band, the fine-scale structure of opal can be obtained.

Sharp bands at 148 cm$^{-1}$ are observed in Opals 01 to 04, which are attributed to anatase [42–44]. Anatase also has bands at 397, 516 and 640 cm$^{-1}$, which can affect the intensity and shape of the broad band at 360 cm$^{-1}$. A small amount of dark anatase inclusions can be observed in those samples with the microscope equipped with a Raman spectrometer. The size of anatase inclusions in our samples is about 1 μm. We subtracted the spectrum of anatase [44] in proportion, and concluded that the impact of anatase on this broad band was very limited. Since our aim was to compare the spectra of different regions on the same sample, and the presence of anatase was not sufficient to affect the comparison of the relative intensity of tridymite and cristobalite type regions, we did not take account of the influence of anatase in the subsequent data analysis.

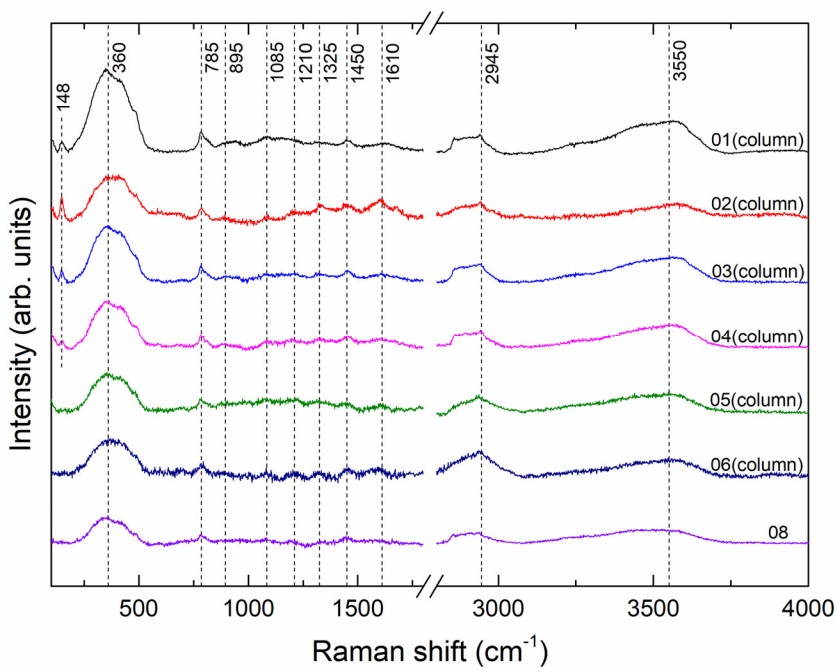

**Figure 9.** Raman spectra in the range of 100–4000 cm$^{-1}$ of opals from Wegel Tena, Ethiopia. Each sample shows a main band at around 360 cm$^{-1}$, several small bands in the range of 750–1650 cm$^{-1}$, as well as broad bands around 2945 cm$^{-1}$ and 3550 cm$^{-1}$. Opals 01 to 04 show a sharp band at 148 cm$^{-1}$, which is related to anatase.

We chose Raman spectra in different areas on the sample for comparison, including (1) The column and matrix at the bottom, and the common opal on the top of Opal-02 (respectively corresponding to "1", "2" and "3" in Figure 3); (2) The column and matrix on Opal-04. Raman spectra data in the range of 170 to 600 cm$^{-1}$ were fitted with five Voigt functions to understand the structure state of the samples (Figure 10). Band parameters after fitting are listed in Table 2.

**Table 2.** Band parameters after fitting the Raman spectra.

| Test Position | R1 [1] | | | R2 | | | R3 | | | R4 | | | R5 | | | $\eta$ [5] | SE [6] $_\eta$ | $\xi$ [7] | SE$_\xi$ |
|---|---|---|---|---|---|---|---|---|---|---|---|---|---|---|---|---|---|---|---|
| | $\omega$ [2] | I [3] | $\Gamma$ [4] | $\omega$ | I | $\Gamma$ | $\omega$ | I | $\Gamma$ | $\omega$ | I | $\Gamma$ | $\omega$ | I | $\Gamma$ | | | | |
| 02 column (bottom) | 230.43 | 4514 | 76.12 | 300.15 | 65467 | 82.96 | 349.95 | 69756 | 74.91 | 415.73 | 83556 | 90.42 | 486.38 | 37986 | 70.49 | 0.61 | 0.14 | 0.83 | 0.17 |
| 02 matrix (bottom) | 233.20 | 3445 | 44.26 | 302.20 | 62437 | 79.54 | 355.50 | 64916 | 70.84 | 418.00 | 76706 | 76.50 | 484.40 | 29522 | 63.40 | 0.61 | 0.17 | 0.93 | 0.20 |
| 02 matrix (top) | 230.44 | 4537 | 124.55 | 307.72 | 36506 | 83.80 | 356.94 | 30301 | 64.80 | 419.37 | 33122 | 74.42 | 486.38 | 20970 | 58.32 | 0.64 | 0.25 | 0.87 | 0.20 |
| 04 column | 237.41 | 12841 | 64.33 | 302.43 | 56312 | 74.68 | 353.50 | 59810 | 64.00 | 416.36 | 91546 | 88.38 | 486.38 | 31884 | 56.43 | 0.53 | 0.09 | 0.72 | 0.09 |
| 04 matrix | 231.13 | 1862 | 28.77 | 298.39 | 49522 | 76.80 | 351.78 | 67882 | 72.95 | 421.38 | 3543 | 90.00 | 490.28 | 15814 | 38.66 | 0.61 | 0.06 | 0.81 | 0.06 |

[1] R1, R2, R3, R4 and R5 refer to the bands at 230, 300, 355, 420 and 490 cm$^{-1}$, respectively. [2] Band position (cm$^{-1}$). [3] Integrated intensity. [4] Full width at half maximum (cm$^{-1}$). [5] Equation (1). [6] Standard Error. [7] Equation (2).

The value of $\eta$ is used to estimate the tridymitic fraction in microcrystalline opal [12], and the calculation formula (Equation (1)) is as follows:

$$\eta = \frac{I(R2) + I(R3)}{I(R1) + I(R2) + I(R3) + I(R4)}, \tag{1}$$

where $I$ is the integrated intensity, and R1, R2, R3, R4 and R5 represent the Raman signals near 230, 300, 355, 420 and 490 cm$^{-1}$, respectively.

For Opal-02, $\eta$ of the matrix (common opal) on the top is larger than the $\eta$ of the column and the matrix at the bottom. Similarly, $\eta$ of the matrix is larger than the columns in Opal-04. This means the tridymite-like structural categories have a larger proportion in the matrix on top compared to their proportion in the matrix and column on the bottom.

The parameter (Equation (2)):

$$\xi = \frac{\Gamma(R3)}{\Gamma(R4)}, \tag{2}$$

where $\Gamma$ is the full width at half maximum (FWHM), and can offer additional information in the structural state. When $\xi$ is closer to 1, the degree of defectiveness in tridymitic and cristobalitic nanoregions will be closer to each other. From column to matrix, the structural defects of tridymite increase relative to cristobalite nanoregions.

The value of $\eta$ is related to the integrated intensity of R1, R2, R3 and R4, and the value of $\xi$ is related to the FWHM of R3 and R4. The estimated standard error calculation formula (Equations (S1) and (S2)) [45] and error bars of $\eta$ and $\xi$ (Figure S1) are shown in the supplementary materials. In this study, each Raman spectrum was only scanned two times for 5 seconds, resulting in a high noise of the spectra, thus affecting the accuracy of the values of $\eta$ and $\xi$. However, the trends are observed, which can support the conclusion. If the acquisition time is prolonged, the spectra with higher quality will be obtained, and the standard error of the values of $\eta$ and $\xi$ will be smaller, which can be more statistically significant and better support the conclusions.

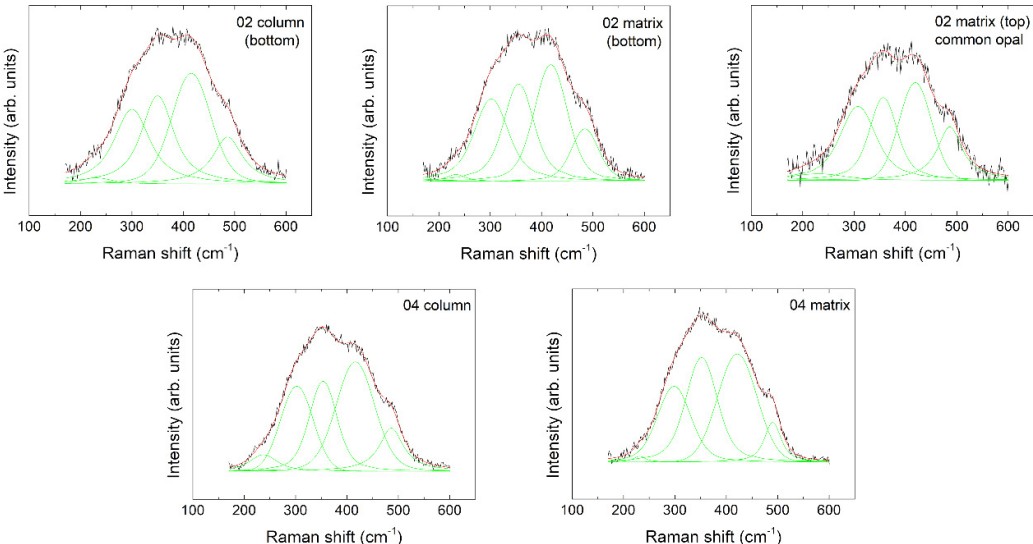

**Figure 10.** Non-linear curve fitting of Raman spectra of different areas in Opal-02 and 04.

### 3.4. Microstructure

### 3.4.1. SEM Observations

All samples consist of aggregates with variable shape, size, ordering and degree of cementation. Silica sphere is a special form of aggregate with a near-spherical shape. On Opal-01, 03 and 04, a layer formed by the combination of silica spheres is observed. Silica spheres are tight-bound, and the gap between them is almost non-existent. The layers are arranged parallel to each other (Figure 11a). The agglomerates formed by the silica spheres are scattered in the layers, with a diameter of about 290 to 350 nm.

Silica aggregates of various shapes (oval, spherical, long stick, etc.; diameter 80 to 500 nm) and a disorderly accumulation are observed on Opal-07 and 08 (Figure 11b). The silica aggregates can make up irregular-shaped agglomerates, making it difficult to observe the gap between the aggregates. Spheres of about 2.2 to 3.8 μm (2200 to 3800 nm) in diameter are observed on the surrounding rock of Opal-07, which is mainly composed of opaque common opal (Figure 11d). We can see a "bridge" of silica linking two neighbouring spheres. This structure was described by Roudeau et al. [14], who proposed that it could be the result of a capillarity phenomenon involving silica deposition from silica-rich water circulation.

In sample Opal-09, where the digit patterns are not well-formed, the boundary between the silica spheres or between the layers is blurred (Figure 11c). The irregularity of the spheres is related to the instability of the geological environment, and most natural opals have this feature. When the hydrothermal environment is affected by high temperatures, the silica spheres are easily agglomerated; when the temperature and pressure are stable, the silica spheres are regularly arranged [46]. We speculate that the environment of Opal-09 was relatively stable at the initial stage of formation, and the silica spheres were arranged in layers. After that, the environment changed, and the original stacked agglomerates or spheres slowly blended together, but the process stopped before it was completed.

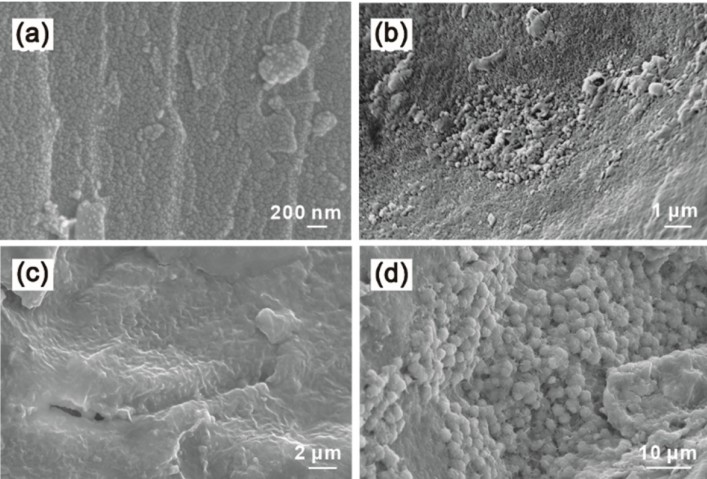

**Figure 11.** Layers and silica aggregates observed by SEM. (**a**) Opal-01 shows a layered structure; (**b**) Silica aggregates of various shapes and agglomerates on Opal-08 (diameters 130–500 nm); (**c**) Blurred boundary of Opal-09; (**d**) Close-packed small spheres on the surrounding rock of Opal-09 with diameters of 2.2–3.8 μm (2200–3800 nm).

### 3.4.2. TEM Observations

Three different types of structural forms are observed in Opal-05 and Opal-06 by TEM. Apparent structure is not observed in Opal-01, because the thickness of the sample does not reach the optimal thickness for a standard TEM observation. The most typical structure is composed of silica spheres in an approximately hexagonal close-packed arrangement, which can be observed in both the

columns and matrix, while the dimensions of the spheres in each area are different (e.g., Opal 05, see Figure 12a,b). The diameters of the spheres in the columns vary from 210 to 240 nm (Figure 12a), while they are about 230 to 260 nm in the matrix (Figure 12b). The electron diffraction pattern is diffuse (Figure 12b), indicating that the spheres are amorphous. Furthermore, two different structures in Opal-06 can also be detected in Figure 12. In the columns, the silica particles are arranged in oriented pyramids of different sizes (Figure 12c), with a width of 840 to 900 nm and a height of 780 to 800 nm. This form is similar to the wedge-shaped twin crystal of tridymite described by Dong [47]. The matrix consists of spheres of varying sizes (250 to 580 nm in diameter) with a distinct boundary (Figure 12d). Parts of the spheres are fused into multiple grains, and the diameter of the grains can be as small as 50 nm.

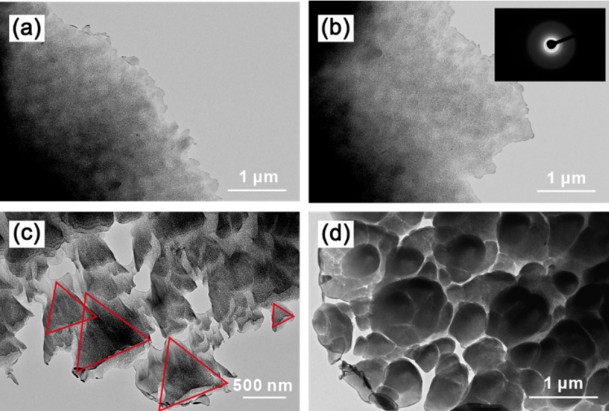

**Figure 12.** Opal observed by TEM. (**a**) Silica spheres in columns (*d* = 210–240 nm) from Opal-05; (**b**) Silica spheres in the matrix (*d* = 230–260 nm) from Opal-05; (**c**) The silica particles in columns from Opal-06 are arranged in pyramids, the width of the cone is about 840–900 nm, and the height is about 780–800 nm; (**d**) Silica spheres in the matrix (*d* = 250–580 nm) with a clear boundary from Opal-06, composed of small spheres, the diameters of which can be as small as 50 nm.

## 4. Discussion

### 4.1. Differences Between Columns and Matrix in Microcrystalline Opal, and Steps of Digit Formation

The Raman spectra indicate that the matrix on the top has a higher tridymite/cristobalite ratio compared to the columns and matrix at the bottom, that means more tridymite structure components will occur closer to the top. In addition, columns have a smaller ξ value than matrix, which means that the degree of order of tridymite structural units increases from the matrix to the columns. At the bottom of the sample, a larger proportion of "tridymite" transforms into "cristobalite", tridymite nanoregions are better ordered in columns, and this state is closer to the end stage of opal-CT formation [12,25,48]. Therefore, we confirm that columns should represent an earlier stage of diagenetic alternation as compared with the matrix, which is consistent with the steps of digit formation [23]. If we consider Opal-02 (Figure 3), the digit pattern is formed in the following steps: (1) Silica spheres undergo sedimentation from the bottom to the top, and then are divided into columns after dendritic; (2) water alters the columns by progressive erosion; (3) new interstitial silica-rich filler settles; (4) additional deposition is covered on top.

### 4.2. The Cause of Play-of-Colour and Iridescence

The consensus for the formation of play-of-colour is that the internal structure of opal is mainly composed of amorphous and microcrystalline silica spheres with similar diameters, which are close-arranged and regular layered [2,13,49]. The distance between the silica spheres is close to the wavelength of visible light, which produces diffraction. There is a direct correlation between the colour, the diameter of the spheres and the distance between them [2,49]. Opal-A always shows a "smooth sphere" structure, while opal-CT shows a "lepisphere" structure [50,51]. Although neither SEM or TEM

observation helped us determine which structure our sample belonged to, the regular arrangement of the spheres was still sufficient to form a diffraction grating that produced play-of-colour.

According to Sanders [51], the wavelength of visible light and the radius of the spheres satisfy the following Equation (3):

$$\lambda_{max} = 2\mu r\sqrt{3}, \tag{3}$$

where $\lambda_{max}$ is the maximum wavelength, $\mu$ the refractive index of opal (1.46) and $r$ the radius of the spheres. The calculated results show that the condition of white light (wavelength ranging from 400 to 760 nm) diffraction can be satisfied when the diameter of spheres is above 158 nm.

If we consider Opal-05 as an example, different networks of play-of-colour can be observed in the columns and matrix (Figure 4). According to Equation (3), since green play-of-colour is observed in the columns, the diameters of spheres in the columns should be in the range of 198 to 228 nm, while when red play-of-colour is observed in the matrix, the diameters of spheres in the matrix should be in the range of 245 to 300 nm. According to the TEM measurements, the diameter of the spheres in the columns is about 210 to 240 nm, and the diameter of the spheres in the matrix is about 230 to 260 nm, which agrees with the calculated result.

The diameter of the silica spheres from the Wegel Tena opal we observed are between 120 and 500 nm, and most of them conform to the calculations of Equation (3). Although the diameters of some silica spheres are extremely small, the diameter of the agglomerates formed by them is about 200 to 580 nm, which conforms to the conditions needed for diffraction, so play-of-colour is possible. Thus, the play-of-colour in Wegel Tena opal can be caused by a three-dimensional diffraction grating formed by silica spheres, including monochromatic diffraction and a layering of play-of-colour that can be shifted from red to yellow to green. The layering of play-of-colour is caused by the gradual change in the silica sphere size—the diameter of the sphere in the red region is larger, and the diameter of the sphere in the green region is smaller. According to [23], the red layer corresponds to the bottom and the green layer to the top. We observed this phenomenon in Opal-01 (Figure 7a). The sedimentation sequence of silica spheres is controlled by gravity, which leads to the deposition of larger spheres first, followed by progressively smaller spheres [23].

In addition to the above two phenomena, Wegel Tena opal can also show rainbow-like continuous spectral colours (Figure 7c), similar to iridescence. Visible interference can occur between the silica layer and the gap between layers. The colour is related to the incident angle of light and the thickness of the silica layer. The interference colour formula for thin-film interference is determined by the following Equation (4) [52]:

$$2\mu h\cos\theta = k\lambda, \tag{4}$$

where $\mu$ is the refractive index of opal (1.46), $h$ is the layer thickness, $\theta$ is the refraction angle, $\lambda$ is the wavelength of visible light (400 to 760 nm) and $k = 1$. $\cos\theta$ decreases in the range of 0° to 90°, so when we take the refraction angle as 0°, we can get the minimum layer thickness of each wavelength in visible light to produce the interference effect. When $\theta = 0°$, the calculated $h$ of 137 to 260 nm can be substituted into the formula. In Opal-07, the diameter of the silica spheres or agglomerates is 120 to 200 nm, and a thin layer (120 to 200 nm thick) fused from a single layer of silica spheres meets the conditions which produce iridescence. As the layer thickness gradually becomes larger or smaller within a small range, monochromatic light of different wavelengths is sequentially strengthened, thereby forming a gradually-changing rainbow-like colour patch observed by the naked eye.

## 5. Conclusions

The opal from Wegel Tena is classified as opal-CT type. The maturity of different regions of the digit pattern can be distinguished by comparing the relative intensity of the Raman signals at around 360 cm$^{-1}$. On the bottom of the sample, a larger proportion of tridymite-type regions transform into cristobalite in the matrix than on the top; while the tridymite-type regions in the matrix contain a higher degree of structural defects than in the columns. In the same digit pattern opal, columns formed earlier than the matrix. The difference in microstructure between the matrix and columns is mainly based on the size or morphology of the silica spheres. The diameter and regular arrangement of the silica spheres in Wegel Tena opals satisfies the conditions for diffraction; as a consequence, the play-of-colour can take place. The silica layer may cause interference of light, resulting in continuous spectral colour patches. Two problems remain to be solved: (1) Whether the opals of the columns and matrices are from the same source; and (2) Determining the elemental differences between the columns and the matrix, and whether this influences the differences in their appearance.

**Supplementary Materials:** The following are available online at www.mdpi.com/2075-163X/10/7/625/s1, Figure S1: Error Bars of the η and ξ values of different areas in Opal-02 and 04, Equation (S1): The formula for estimating the standard error of η, Equation (S2): The formula for estimating the standard error of ξ.

**Author Contributions:** Conceptualisation, K.Z.; methodology, K.Z. and F.B.; validation, K.Z.; formal analysis, K.Z.; investigation, K.Z.; resources, K.Z. and F.B.; data curation, K.Z.; writing—original draft preparation, K.Z.; writing—review and editing, F.B.; visualisation, K.Z.; supervision, F.B.; project administration, F.B. All authors have read and agreed to the published version of the manuscript.

**Funding:** This research received no external funding.

**Acknowledgements:** We thank Opal Style for providing samples from reliable sources. Thanks to the School of Gemmology, Institute of Earth Science, School of Earth Sciences and Resources of the China University of Geosciences, Beijing, and the School of Materials Science and Engineering of the University of Science and Technology, Beijing for the experimental equipment. Thanks to Li Zhechen, Hu Zhikang, Zhang Yu and Sun Zhulin for their help with instrument operation.

**Conflicts of Interest:** The authors declare no conflict of interest.

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
