# Peer review of "Crystallinity and Play-of-Colour in Gem Opal with Digit Patterns from Wegel Tena, Ethiopia"

_minerals, doi:10.3390/min10070625_

Round 1

Reviewer 1 Report

The resubmitted manuscript combines a range of analytical methods in a detailed investigation of gem opal from Wegel Tena. All my concerns and critical questions raised in the previous rounds of the review process have been thoroughly addressed by the authors. Specifically, treatment of the available Raman data has been improved, and results now support the presented conclusions. This valuable contribution will certainly find an interested audience in the opal community and motivates future spectroscopic studies on an extended data set.

Other remarks: final reading should eliminate the density of minor spelling and typographical errors. E.g.:

10: between

11: scanning and transmission electron microscopy

13: spectra

14: indicates

15: a higher amount

20: Raman spectroscopy

25: relatively well-ordered

28: conclusive [1]

81: software.

Table 1: formatting (some words have been broken)

120: opal [23] 

176: 490 cm-1

177: terms [

196: position

242: we thought this is

243: consists of / comprises / made up of

303: gravity, which

322: 360 cm-1; 11: contains

420, 427: SiO2

440: less capital letters

Kind regards

Author Response

Dear Reviewer 1:

Thank you for your continuous attention to our article. Your suggestions helped a lot to our article.

We have corrected all the errors about the minor spelling and typographical you marked.

Table 1 was resized and a new footnote was added to improve the format.

Other  typographical  errors are corrected as follows:

line 46: columns,

line 70: require preparation

line 168: background,

Best Regards

Reviewer 2 Report

Dear  Authors,

I think this is a good paper but I suggest many corrections in the attachment file.

Best regards

Author Response

Dear Reviewer 2:

Thank you for your approval of our article and your thoughtful suggestions. 

Our reply and corrections of manuscript are in the PDF file. 

This manuscript is a resubmission of an earlier submission. The following is a list of the peer review reports and author responses from that submission.

Round 1

Reviewer 1 Report

The work made by K. Zhao and F. Bai  on crystallinity and play of colour in precious opal with digit pattern from Wegel Tena in Ethiopia is a very worthy and  well organized. I read it with great interest. For me the discussion on the cause of play of colour and iridescence effects is especially valuable. The manuscript provides interesting and partly new data concerning the micro-structure of   precious opal with digit pattern and is suitable for a publication in the Minerals.  However, there are a few shortcomings, which should be revised in a revised version of the manuscript:

  1. In all FTIR spectra collected from opals samples the band at ca. 780 cm-1 has very low intensity. Definitely it is not strong band, as authors mentioned in the text and figure caption of Fig. 8. Moreover it is very broad, so it is not easy to find the maximum of such band. In this situation, the position of the band attributed to the symmetric stretching vibration of Si-O could not be treated as conclusive in distinguishing opal CT from opal A. The better results in distinguishing opal A from opal CT, and Opal C could be derived from powder XRD.
  2. In the paragraph ‘ Materials and methods’ authors mentioned that 3 opal samples (no 1, 5, 6) were observed under TEM. In the results only 2 opals (no 5 and 6) were described. As a result three structural forms were recognized: regular-arranged silica spheres, silica forming pyramids, and spheres with distinct boundaries between them. Which structural type of silica was observed in sample opal 1?

Minor mistakes or vagueness of some statement as well as some typos, which I found  I marked in the pdf file.

Reviewer 2 Report

To provide a detailed investigation on a specific type of hydrous silica, the submitted manuscript on “Crystallinity and play-of-colour in gem opal with digit patterns from Wegel Tena, Ethiopia” combines results from optical microscopy, spectroscopy (IR and Raman), and electron microscopy (SEM and TEM). Besides a general description, distinct domains composing a digit pattern are characterized and discussed within the framework of a previously proposed formation model (Rondeau et al. 2013, Gems & Gemology). The addressed topic and analytical approach would be perfectly suited for publication in a mineralogical journal. As yet, however, the existing Raman data (i.e. the scientific core of the presented study), is flawed in data acquisition, processing, and interpretation. Accordingly, central conclusions concerning silica crystallinity are not well supported (detailed comments below).

1.) Raman data acquisition (Fig. 9, Fig. 10)

  • Several spectra (Opal-05, 06, 08) suffer from pronounced artifacts. Due to fluorescence effects, decreasing over the time of laser irradiation, the detected intensity dropped when the acquisition window was moved. Although not detrimental for the observation of most of the spectral features, that’s not the way how Raman data should be recorded, and it’s certainly not adequate for a publication. High quality silica data could be acquired after some time delay to “burn out” fluorescence, increased overlap and activated intensity adjustment of the acquisition windows, or by using the continuous scanning mode.
  • The spectra are unnecessarily high in noise, considering that the aim was to detect minute differences in silica crystallinity or order. Acquisition time (given as 2 x 5 seconds) should be increased, and confocal hole could be opened. Description of experimental conditions is missing other important information (such as laser power, grating, slit, calibration).

2.) Raman data processing (Fig. 9, Table 2, Fig. 10, Table 3)

  • No information is given on the fitting routine for the data presented in Table 2. Anyhow, peak positions listed in Table 2 significantly differ from peak positions listed in Table 3 for the same material (e.g. Opal-01: 349 vs. 373 cm^-1, 783 vs. 788-790 cm^-1). Authors should prepare a consistent data set.
  • There is an unassigned band in the spectra of opal-01 to -04. Figures are missing the tick marks on the x-axis. However, a band at ~ 140 -145 cm^-1 might be due to anatase, for instance (RRUFF data base). If anatase is present as an additional phase, it would also contribute additional bands as interferences with the main opal-CT peaks of interest. At any rate, these spectra have not been obtained on a pure silica phase. The additional band must be assigned and potential contributions of interferences must be evaluated.
  • The authors fitted one Gaussian functions to each of the two main peaks of interest. This is definitely not the way of Raman data processing to extract information on silica crystallinity and structural order. The broad peak at ~ 350 cm^-1 is a composite peak, consisting of - at least - 4 discrete bands (originating from cristobalite and tridymite), and potentially also interferences from the unassigned phase. Similar, there are two bands contributing to the peak at ~ 780 cm^-1. The authors have cited the authoritative Raman study of Ilieva et al. (2007, Am. Min.) on opal-CT; data treatment must involve peak deconvolution (and using Gaussian-Lorentzian functions), to extract crystallinity information.

3.) Raman data interpretation

  • Contrary to the authors’ interpretation, the FWHM of a composite peak, consisting of at least 4 distinct silica bands, bears no information on the domain size (as a measure of crystallinity). Main peak at ~ 350 cm^-1 would broaden with decreasing tridymite/ cristobalite ratio. The domain size information can only be obtained from peak deconvolution and the FWHM of the individual bands (see Ilieva et al. 2007). Contrary to the authors’ statement, differences in crystallinity (here: domain size) have not been demonstrated.
  • In the discussion section, the present text gives the impression, as if size of silica spheres observed by TEM (e.g. 210 - 260 nm) would relate to domain size obtained by Raman spectroscopy. This is not the case and should be specified. Ilieva et al. (2007) demonstrated domain size effects (again, for the deconvoluted Raman bands!) for domain sizes of 7 to 21 nm (from powder XRD). Wegel Tena opal might typically have domain sizes around 6 nm (as determined from powder XRD, e.g. Liesegang & Tomaschek 2020, Sed. Geol.).
  • The degree of structural order (tridymite/ cristobalite ratio) has to be obtained from the intensity ratios of the respective bands contributing to the main peak at ~ 350 cm^-1 (see Ilieva et al. 2007). From optical inspection of the data in Figure 10, the matrix seems to be lower in this ratio, has an increased amount of cristobalite, is higher in structural order (or maturity). This would be in contrast to the assumptions and conclusions presented by the authors: “It is reasonable to assume that that the degree of structural order is relatively high for the area earlier formed [...], therefore we confer that the formation of columns is earlier than the matrix”.
  • The present manuscript does not present Raman data in support of a main conclusion or the proposed formation model. I would highly recommend that the authors improve the steps of Raman data acquisition, processing, and interpretation.

I enjoyed learning something more about the exciting topic addressed in this manuscript. Thanks & kind regards.

Round 2

Reviewer 2 Report

It’s a pleasure to read that the authors have revisited the available Raman data, thoroughly addressed critical issues, and now provide a much better presentation that significantly improved the manuscript. Specifically, the authors have applied the model of Ilieva et al. (2007) to extract a structural characterization of the investigated opal-CT from Raman spectroscopic data. The presented results, however, still do NOT support inferences relating to opal maturity.

Ilieva et al. suggested that Raman scattering of opal-CT in the ~ 350 cm^-1 region can be deconvoluted into four separate bands, with positions that are consistent with those of tridymite and cristobalite, respectively (231, 304, 351, 418 cm^-1). Relative intensities of these bands could then be used to assess the relative amount of tridymite and cristobalite structural units in opal-CT. Although this simple model had also been questioned (Ivanov et al. 2011, J Phys Chem C), it seems well established and applied for opal-CT characterization (see literature citing Ilieva et al. 2007).

Now, back to the data presented in the manuscript: The authors have fitted four Voigt functions to the ~ 350 cm^-1 region. The text (and Table 2) lists the resulting band positions as 230, 310, 360 and 440 cm^-1. Whereas the 3 positions at lower wave number seem broadly consistent with the band positions expected for tridymite and cristobalite, the band position at ~ 440 (435-445) cm^-1 is off by ~20 wavenumbers from a cristobalite band at 418 cm^-1. Accordingly, results of this fit are certainly NOT consistent with a simple binary mixing model, intensity ratios do NOT represent the relative proportion of cristobalite and tridymite structural units, and do NOT allow for conclusions about a sequence of silica maturation in the distinct domains.

From expecting the spectra shown in Fig. 10, it may be clear, why an uncritical application of the model failed to appropriately describe the presented data. The spectra show a sub-maximum at ~ 418 cm^-1 and a pronounced shoulder at ~ 490 cm^-1. The curve at ~ 440 m^-1 of the applied four-band-fit integrates over both regions. I.e., it integrates over what may represent a cristobalite band (418 cm^-1) and an additional contribution at ~ 490 cm^-1. It seems that, in contrast to the spectra presented by Ilieva et al. (2007), the additional contribution can’t be neglected here.

Not to offer a final solution, but some suggestions: Take into account the additional ~ 490 band in a fit to extract the tridymite and cristobalite contributions. Provide a rational for the additional band: what does it represent, is it a sample property (e.g. also present in published spectra from Wegel Tena opal?) or an analytical artifact?

Revisited Raman data may finally support the suggested maturation sequence. I’m certainly not calling for an extended Raman study (although it would be worth for this interesting material), just for consistency of the presented data and conclusions.

Other remarks:

line 64: polishing have has no effect

line 80ff: what is the output power of the 532 nm laser? (Important, since high in laser energy will destroy the silica sample and possibly introduce artifacts). Some calibration routine for wave number was applied?

line 291: matrix has a lower higher tridymite/cristobalite ratio compare compared to columns

throughout the Raman text: “peaks” may better be replaced by “bands”

references # 4, #41: use lowercase letters

Kind regards